# Monitoring Compositional Changes in Black Soldier Fly Larvae (BSFL) Sourced from Different Waste Stream Diets Using Attenuated Total Reflectance Mid Infrared Spectroscopy and Chemometrics

**DOI:** 10.3390/molecules27217500

**Published:** 2022-11-03

**Authors:** Louwrens C. Hoffman, Shuxin Zhang, Shanmugam Alagappan, Volant Wills, Olympia Yarger, Daniel Cozzolino

**Affiliations:** 1Centre for Nutrition and Food Sciences, Queensland Alliance for Agriculture and Food Innovation (QAAFI), The University of Queensland, Brisbane, QLD 4072, Australia; 2Fight Food Waste Cooperative Research Centre, Wine Innovation Central Building Level 1, Waite Campus, Adelaide, SA 5064, Australia; 3Department of Animal Sciences, University of Stellenbosch, Private Bag X1, Matieland, Stellenbosch 7602, South Africa; 4School of Agriculture and Food Sciences, Faculty of Science, University of Queensland, Brisbane, QLD 4072, Australia; 5Goterra, 14 Arnott Street, Hume, Canberra, ACT 2620, Australia

**Keywords:** black solider fly, infrared, instar, waste, chemometrics

## Abstract

Black soldier fly (*Hermetia illucens*, L.) larvae are characterized by their ability to convert a variety of organic matter from food waste into a sustainable source of food (e.g., protein). This study aimed to evaluate the use of attenuated total reflectance (ATR) mid-infrared (MIR) spectroscopy to monitor changes in the composition as well as to classify black soldier fly larvae (BSFL) samples collected from two growth stages (fifth and sixth instar) and two waste stream diets (bread and vegetables, soy waste). The BSFL samples were fed on either a soy or bread-vegetable mix waste in a control environment (temperature 25 °C, and humidity 70%). The frass and BSFL samples harvested as fifth and sixth instar samples were analyzed using an ATR-MIR instrument where frequencies at specific wavenumbers were compared and evaluated using different chemometric techniques. The PLS regression models yield a coefficient of determination in cross-validation (R^2^) > 0.80 for the prediction of the type of waste used as diet. The results of this study also indicated that the ratio between the absorbances corresponding to the amide group (1635 cm^−1^) and lipids (2921 + 2849 cm^−1^) region was higher in diets containing a high proportion of carbohydrates (e.g., bread-vegetable mix) compared with the soy waste diet. This study demonstrated the ability of MIR spectroscopy to classify BSFL instar samples according to the type of waste stream used as a diet. Overall, ATR-MIR spectroscopy has shown potential to be used as tool to evaluate and monitor the development and growth of BSFL. The utilization of MIR spectroscopy will allow for the development of traceability systems for BSFL. These tools will aid in risk evaluation and the identification of hazards associated with the process, thereby assisting in improving the safety and quality of BSFL intended to be used by the animal feed industry.

## 1. Introduction

According to the United Nations, the global population will reach 9.4–10.0 billion in 2050 [1]. Current agricultural production practices have placed pressure on natural resources causing significant environmental impacts (e.g., loss of nutrients) [2,3,4,5,6,7,8]. The water and land footprint of the current and conventional agricultural production systems is driving a change towards more viable and sustainable systems. It has been reported that more than 1.3 billion tons of food produced for human consumption is wasted annually, while the management of these waste streams through landfills and incineration also have a negative impact on the environment [2,3,4,5,6,7,8]. Therefore, alternatives to manage or utilize waste in a sustainable way have been proposed.

Adding value to food waste using insects such as black soldier fly larvae (BSFL) has been receiving attention in the last 20 years [9,10,11,12,13,14,15]. The black soldier fly (*Hermetia illucens*) belongs to the family Stratiomyidae, a kind of true fly with adults being 13–20 mm in length [5,6,7,8,9]. This fly originated from the Americas but is now found in most tropical and subtropical regions [9,10,11,12,13,14,15]. This insect prefers to lay eggs in warm conditions (oviposition almost always occurring at temperatures > 26 °C), whilst the diversity and the efficiency of substrate digestion by the larvae are probably the highest compared with other species of flies [9,10,11,12,13,14,15]. What makes BSF an ideal species for the upcycling of waste is not only the efficiency of the BSFL in utilizing the waste, but the fact that the adult flies do not eat-rather, they only drink some fluid, thus minimizing their ‘nuisance’ factor compared to other fly species such as the domestic fly. There is also the added advantage that the waste residue, called frass, has the potential for utilization as a soil enumerator [16].

The larvae of BSF go through six larval instars [9,10,11,12,13,14,15]. These larvae are polyphagous saprophagous that can actively forage on a variety of organic matter sources [8,9,15,16]. The duration of the larval stages can range between 10 and 52 days depending upon the type of substrate, temperature, and humidity of the feeding environment [8,9,15,16,17]. The larvae in the prepupa stage comprise 42% protein and 35% lipid, including essential amino acids and fatty acids [8,9,15,16,17]. The growth efficiency, ability to biodegrade organic matter, and the composition of the larvae can be considerably influenced by the substrate (e.g., type and composition of the diet used) [8,9,15,16,17].

Routine methods used to monitor and analyze the chemical composition of the BSFL are based on the use of the proximate analysis and include the analysis and evaluation of protein, fat, ash, and moisture contents [8]. However, these methods are time consuming and expensive to be used by the industry. Therefore, techniques based on infrared (IR) radiation have become a cost-effective alternative as they are considered green, non-destructive, and high-throughput methods [18,19,20,21].

Infrared spectroscopy is a well-known analytical technique that provides quantitative chemical, compositional, and functional information about the sample [18,19,20,21]. This technique measures the absorption of IR radiation from chemical bonds within the sample [19,20,21]. The spectral peaks derived from the functional groups in the mid-infrared (MIR) range are sharper having higher resolution than those observed in the near-infrared region (NIR) [17,18,19,20]. In addition, the MIR region provides the so-called fingerprint region that defines and records the intrinsic characteristic of a compound or sample [18,19,20,21]. Thus, this information can be used to characterize the chemical composition of the samples or to monitor changes during the process [18,19,20,21].

This study aims to evaluate the ability of using attenuated total reflectance (ATR) MIR spectroscopy to monitor the composition of BSFL samples (5th and 6th instars and frass samples) as well as to classify samples reared using two different waste stream diets (bread-vegetable mixtures and soy waste).

## 2. Results and Discussion

The ATR-MIR average spectra of the BSFL samples collected at the 5th and 6th instar as well as the frass samples analyzed are shown in Figure 1 (both waste streams combined). The MIR spectra of the BSFL larvae samples collected at the two instars showed differences at specific frequencies or wavenumbers. A broadband can be observed around 3000–3500 cm^−1^ associated with both the O-H and N-H stretching bands related to the water and protein content of the samples as reported by others [22,23,24]. In addition, two peaks can be observed around 2853 cm^−1^ and 2919 cm^−1^ associated with lipid content. These peaks might also be associated with the presence of saturated fatty acids in the BSFL samples [22,23,24]. The observed changes in the MIR absorbance values at these frequencies can be associated with the developmental stages of the larvae, as the lipid composition changes during larvae development [25,26]. The average MIR spectra of the frass samples analyzed indicated that the absorbance values at frequencies corresponding to both lipids and protein were lower than those observed for the instar samples. In addition, the frass samples showed high absorbance values around 1020 cm^−1^ associated with C-H linked to polysaccharides (e.g., starch) and other carbohydrates [22,23,24,27].

High absorbance values were observed in the MIR spectra of the 6th instar samples analyzed around 1635 cm^−1^, 1571 cm^−1^, and 1538 cm^−1^ associated with amide II, related with C=O and N-H stretching bands. For the 5th instar samples, the high absorbances associated with protein were observed around 1628 cm^−1^ and 1536 cm^−1^. These peaks have been reported to be associated with both protein and chitin content [27,28,29]. It has also been reported by other authors analyzing BSFL samples that peaks associated with chitin can be found around 1632 cm^−1^, 1621 cm^−1^, and 1538 cm^−1^ [27,28,29]. The same authors have also indicated that α-chitin has three characteristic peaks in the MIR region between 1660 cm^−1^ and 1625 cm^−1^, while the presence of β-chitin can be observed around 1656 cm^−1^, associated with the C=O second stretch of amide I, and N-H and C-N stretching of the amide II [27,28,29]. According to these authors, chitosan can be associated with the absorbances around the 1590 cm^−1^ wavenumber. It is well known that during larvae development, the content of chitin changes, and more specifically the content of chitin decreases from the larvae stages to the prepupa stage [26,30,31,32]. An increase in the absorbance values around 1240 cm^−1^ associated with polysaccharides was also observed in the samples collected from the 6th instar.

It has been reported that the larvae composition varies depending on the waste stream used, where lipids, ash, and fibre are the most prevalent macronutrients [30,31]. To evaluate the effect of the two waste stream diets utilized, the ratio of the absorbances corresponding to the amide I (1634 cm^−1^) to amide II (1528 cm^−1^) groups (Figure 2A) and the ratio between the amide groups and lipids (2921 + 2849 cm^−1^) (Figure 2B) were calculated for both types of waste or diets utilized to rear the larvae. These ratios represent the changes in the composition of the larvae associated with protein and lipid composition related to either the development stages of the larvae (instar) or the diet used. It was observed that the ratio (1634/1528 cm^−1^) was higher in the 5th instar than in the 6th instar samples. This ratio was also higher in the instar samples fed with the bread-vegetable mix waste diet compared with that fed using the soy waste diet (see Figure 2A,B). The ratio for the absorbance associated with the protein–lipid composition has the same trend as the one observed for the amide groups. It has been also reported that BSFL fed with diets with a high carbohydrate content tend to have more lipids [26,30,31,32]. The results of this study showed a similar trend as reported by these previous authors [26,30,31,32].

The principal component score plot of the combined instars (5th and 6th) and frass samples fed using either the soy or bread-vegetable mix is shown in Figure 3A. The first four principal components explained >90% of the variance in the dataset. A separation was observed corresponding to the type of samples (instar vs frass) along PC1 (49%), whilst a separation between instars was observed along PC4 (5%). The separation between the samples according to the type of waste stream was clear in the frass samples analyzed (see Figure 3B). However, some overlap was observed in the instar samples; thus, a separate PCA analysis was carried out to specifically analyze the effect of the type of waste stream in each instar. In this study, a separate PCA was carried out using only the 6th instar samples. The principal component score plot and loadings for the analysis of the BSFL samples collected from the 6th instar are shown in Figure 4 (score plot and loadings). The PCA score plot (PC2 vs PC4) showed a separation between the 6th instar samples according to the type of waste stream along PC4 (2%). The highest loadings were observed at 1546 cm^−1^, 1482 cm^−1^, and 1395 cm^−1^. These peaks are associated with the amide groups, nitrogen compounds, and chitin, mainly associated with the high content of protein in soy waste compared with the bread–vegetables mix.

A PLS discriminant regression analysis was carried out using the region between 3000 and 2500 cm^−1^, the fingerprint region, and the full MIR range (see Table 1) [17,32]. The coefficients of determination (R^2^) and standard error in the cross-validation were 0.90 (SECV: 0.16), 0.90 (SECV: 0.16), and 0.90 (SECV: 020) using the fingerprint, lipid, and full MIR ranges, respectively. In all cases, the samples were 100% correctly classified according to the type of waste used (see Table 1). The coefficients of regression using the fingerprint region showed 1595 cm^−1^ and 1484 cm^−1^ associated with amide and nitrogen, respectively [18,33]. Using the lipid region, the highest loadings were observed around 2905 cm^−1^ and 2855 cm^−1^ [18,33]. Using either the lipid region or the fingerprint region, the samples were 100% correctly classified.

## 3. Material and Methods

### 3.1. Experimental Protocol, Samples, and Sampling

The BSFL samples were sourced from two food waste stream diets (namely soy waste and bread-vegetable mix) under commercial conditions. Ten thousand BSF eggs hatched to 5-day-old (chronological age) BSFL were added into a tray (60 × 40 cm) containing 2 kg of feed. The experimental design consisted of two diets (soy or bread-vegetable mix) with six trays per diet (n = 12; six per diet). The bread and vegetable mixture waste consisted of 60% *f*/*w* bread and 40% *f*/*w* fruits and vegetables. The larvae were fed ad libitum and monitored regularly. When the first pre-pupae and pupae were observed, the 5th and 6th instar larvae were harvested from each of the trays. Approximately 50 g of larvae per tray was randomly collected and selected from the 5th and 6th instar larvae. Then, the samples were homogenized. In addition, 50 g of frass from each tray was collected after harvesting the instars. The moisture in the trays was maintained above 70% whilst the temperature of the rearing room was maintained above 25 °C. Details of the experimental protocol and design can be found elsewhere [8].

### 3.2. Mid-Infrared Spectra Collection

Samples (5th (n: 7) and 6th instars (n: 24) and frass (n: 36)) were analyzed using a platinum diamond attenuated total reflectance (ATR) cell attached to a Bruker Alpha MIR instrument (Bruker Optics GmbH, Ettlingen, Germany). The BSFL larvae and frass samples were placed on top of the ATR diamond cell and scanned once in the MIR range (4000 to 400 cm^−1^) where the sample spectrum was the result of twenty-four coadded spectra with a resolution of 4 cm^−1^. The software OPUS version 8.5 provided by Bruker Optics (Bruker Optics GmbH) was used to record the MIR data and to control the instrument (e.g., diagnostics). Air was used as a background and the ATR crystal was cleaned with ethanol and water (70% *v*/*v*) after each sample.

### 3.3. Data Analysis

The data analysis was performed using the Unscrambler X software (v11, CAMO ASA, Oslo, Norway). The MIR spectra were first preprocessed using the baseline and the Savitzky-Golay second derivative (second-order polynomial and a smoothing window size of 10 points) [34]. Principal component analysis (PCA) was performed to visualize the data structure and identify trends as well as the presence of outliers. Partial least squares regression analysis (PLS-DA) was achieved using the MIR spectra (X variables) and a dummy value corresponding to the type of waste stream [35]. The PCA and PLS-DA models were developed using full cross-validation (leave one out). The optimal number of factors for the calibration model was selected based on the minimal value of the predicted residual sum of squares (PRESS) that provided the highest correlation coefficient (R^2^) between the actual and predicted values [35]. The PLS models were evaluated using the standard error of cross-validation (SECV), and coefficient of determination in cross validation [35]. The per cent of correct classification was also used to evaluate the PLS-DA regression models. 

## 4. Conclusions

The results of this study indicate that the ratio between the absorbances corresponding to the amide group and lipid region was higher in waste streams containing a high proportion of carbohydrates (bread-vegetable mix diet). The results of this study further demonstrate the ability of MIR spectroscopy combined with chemometrics to monitor and classify BSFL instar samples according to the waste stream used as a diet. Further studies should be carried out to evaluate the ability of MIR spectroscopy to trace BSF larvae when reared using other organic waste streams such as municipal and abattoir waste as well as different heterogenous matrices (e.g., the combination of hospital and other food wastes). Overall, MIR spectroscopy has shown potential to be used as a tool to evaluate and monitor the development, growth, and type of waste stream used to feed BSFL. The utilization of MIR spectroscopy will allow for the development of traceability systems for BSFL. These tools will aid in risk evaluation and the identification of hazards associated with the process, thereby assisting in improving the safety and quality of BSFL intended to be used by the animal feed industry.

## Figures and Tables

**Figure 1 molecules-27-07500-f001:**
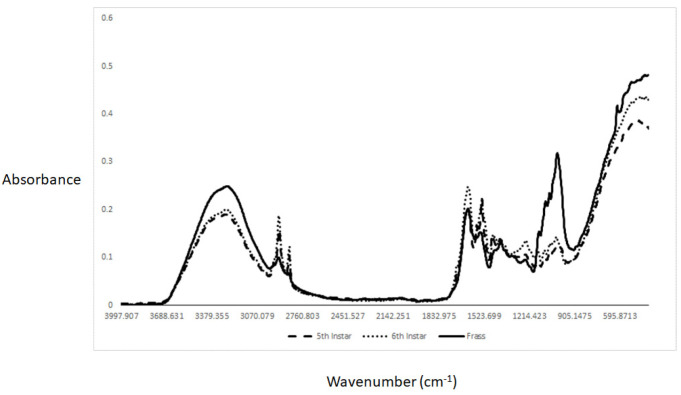
Average spectra (both waste streams combined) of instars (5th and 6th) and frass samples analyzed using attenuated total reflectance mid-infrared spectroscopy after baseline correction.

**Figure 2 molecules-27-07500-f002:**
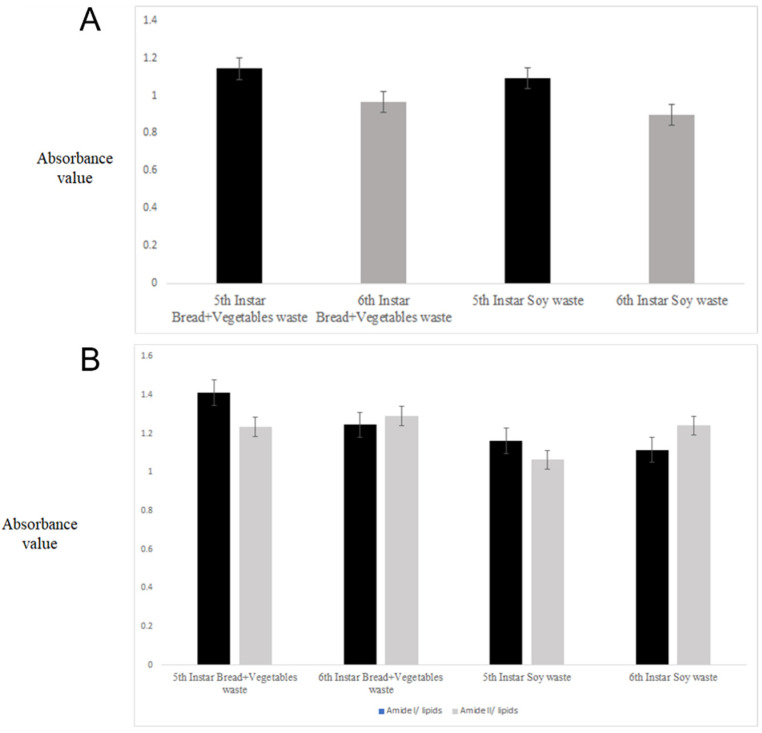
Ratios between the different absorbance values derived from the analysis of different instars (5th and 6th) and type of waste. Panel (**A**) is the ratio between the 1650 and 1540 cm^−1^ frequencies; panel (**B**) is the ratio between the 1650 and 1540 cm^−1^ frequencies and lipids (2921 + 2849 cm^−1^) range.

**Figure 3 molecules-27-07500-f003:**
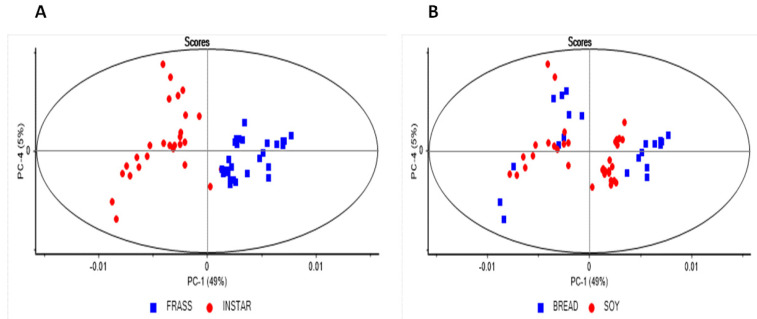
Principal component score of the combined set of instars (5th and 6th), frass samples, and type of waste (diet) analyzed using attenuated total reflectance mid-infrared spectroscopy. Panel (**A**) samples marked by source (instar vs frass); panel (**B**) samples marked by type of waste stream.

**Figure 4 molecules-27-07500-f004:**
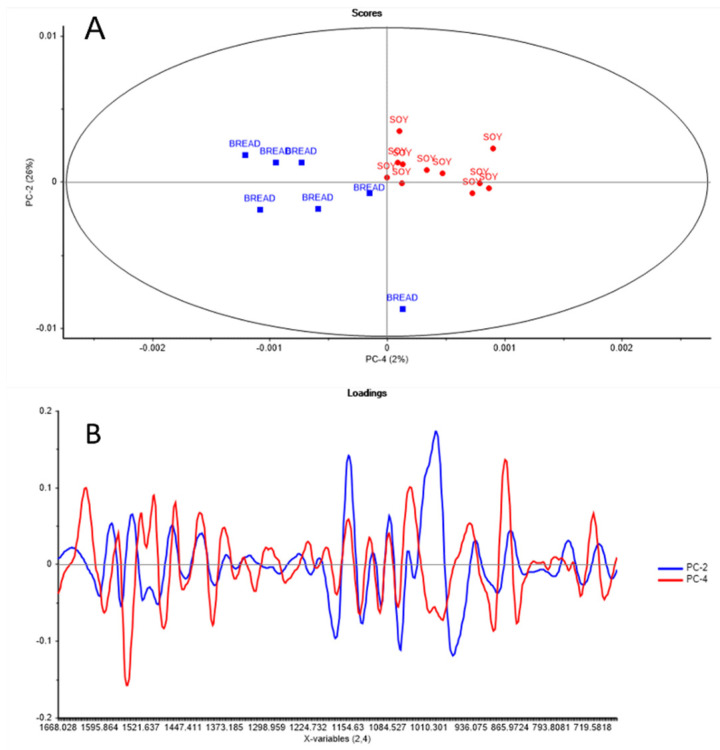
Principal component score plot (**A**) and loadings (**B**) of the 6th instar samples sourced from the two waste streams (bread and vegetables; soy) analyzed using attenuated total reflectance mid-infrared spectroscopy.

**Table 1 molecules-27-07500-t001:** Statistics for the cross-validation prediction of the waste stream diets in the combined set of 5th and 6th instar samples analyzed using attenuated total reflectance mid-infrared spectroscopy.

	R^2^	SECV	LV
Fingerprint region (1800 to 500 cm^−1^)	0.85	0.19	4
Lipid region 3000 to 2500 cm^−1^	0.90	0.16	7
Full MIR range (3000–500 cm^−1^)	0.90	0.20	4

R^2^: coefficient of determination; SECV: standard error in cross validation; LV: latent variables.

## Data Availability

Not applicable.

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
