# Peer review of "Monitoring Compositional Changes in Black Soldier Fly Larvae (BSFL) Sourced from Different Waste Stream Diets Using Attenuated Total Reflectance Mid Infrared Spectroscopy and Chemometrics"

_molecules, 2022, doi:10.3390/molecules27217500_

Round 1

Reviewer 1 Report

This study aimed to evaluate the use of attenuated total reflectance (ATR) mid-infrared (MIR) spectroscopy to monitor changes in the composition of black soldier fly larvae (BSFL) collected from two growth stages (5th and 6th instar) and two waste stream diets. Obtained some good results, which is useful for the study of some insects.

Comments

1. line 116-117, for the classification work, the PLS-DA was used. Usually, the identification accuracy was used to evaluate the calibration performance for a qualitative problem, however, the PRESS, R2, and SECV, which are used for the evaluation of quantitative calibration performance, were used in this work. Was it reasonable?

2. Line 178. “a PCA was carried out using samples collected as 6th instar.”, The authors should give the results of that of 5the instar.  

3. For table 1. The number of LV has a great influence on the calibration performance. For, Lipid region 3000 to 2500 cm-1, 4 LV should be used ( as same as the others), then a reasonable result can be obtained.

Author Response

Dear Reviewer:

Thank you for your comments and suggestions. Below are the answers in bold.

Rev 1

This study aimed to evaluate the use of attenuated total reflectance (ATR) mid-infrared (MIR) spectroscopy to monitor changes in the composition of black soldier fly larvae (BSFL) collected from two growth stages (5th and 6th instar) and two waste stream diets. Obtained some good results, which is useful for the study of some insects.

Comments

  1. line 116-117, for the classification work, the PLS-DA was used. Usually, the identification accuracy was used to evaluate the calibration performance for a qualitative problem, however, the PRESS, R2, and SECV, which are used for the evaluation of quantitative calibration performance, were used in this work. Was it reasonable? Thank you for the comments. We have used PLS-DA to predict the waste stream, not to develop a classification model as such.  In the original draft we have included a sentence that said, “In all cases, samples were 100% correctly classified according to the type of waste used”. We have corrected and clarified in the text.

  1. Line 178. “a PCA was carried out using samples collected as 6th instar.”, The authors should give the results of that of 5the instar.   Due to sampling issues, we did not have enough samples from the 5th instar stage to develop a proper PCA. For this reason, we did not report the PCA for the 5th instar.

  1. For table 1. The number of LV has a great influence on the calibration performance. For, Lipid region 3000 to 2500 cm-1, 4 LV should be used ( as same as the others), then a reasonable result can be obtained. The number of LV was defined by the software, and it was calculated by the PRESS function and cross validation. This is the reason of having different LV.

Reviewer 2 Report

Abstract should be extended with the key results obtained for chemometrics (PCA, PLS models, evaluated using cross-validation).

Regarding Introduction part, the aim of the study (last paragraph) should be extended: please clearly describe why did you use Principle Component Analysis and PLS-DA. Please emphasize why is the combination of spectra scanned in the MIR range, together with chemometric tools, important for your research.

Regarding materials and methods: Subchapter 2.1. Experimental protocol should be clearly described. Is there known methodology (from literature) for performing described experimental design ? If so, please add reference.

Subchapter 2.2. You mentioned that the sample spectrum was the result of  24 coadded spectra with a resolution of 4 cm-1. Why 24 ? Please specify number of samples during sampling period (5th instar, 6th instar, frass) and how many spectra did you record per sample ?

Subchapter 2.3. This subchapter should also be clearly described. If I understood correctly, preprocessed MIR spectra (baseline and Savitzky-Golay) were further used for performing PCA and PLS-DA ? It would be easier for following if you describe, in one sentence, the reason of PCA performing and in one sentence why did you choose PLS-DA ? Applicability of PLS models developed using preprocessed spectra was estimated based on the standard error of cross-validation and coefficient of determination in cross validation. What about RPD (residual predictive deviation) in order to evaluate a predictive ability of a given model ? Please calculate RPD.

Results and discussion: Caption of figure 1 should contain type of waste. It is unclear is it for soy or vegetable mixture waste. Also, you should add the word preprocessed (average preprocessed spectra). You did not present figures of MIR spectra for both type of waste. Why ?

Paragraph describing ratio of the absorbance between specific groups (panel A and B) should be clearly described for each type of waste. Accordingly, caption of figure 2 should be improved.

Descriptions of figures 3 and 4 should be extended. For instance, figure 3, ....the combined set of instar (5th and 6th) and frass samples. For which type of waste ? Figure 4: for which type of waste ?

Description of table 1: ....the waste stream diet.....(for which type of waste) ?

Conclusion should be rewritten in a more scientific manner. The last sentence in a conclusion is unclear.

Author Response

Dear Reviewer:

Thank you for your comments and suggestions. Below are the answers in bold.

Abstract should be extended with the key results obtained for chemometrics (PCA, PLS models, evaluated using cross-validation). We have added these results in the Abstract as suggested by the reviewer.

Regarding Introduction part, the aim of the study (last paragraph) should be extended: please clearly describe why did you use Principle Component Analysis and PLS-DA. Please emphasize why is the combination of spectra scanned in the MIR range, together with chemometric tools, important for your research.  The reasons for using PCA, PLS and MIR are already explained and detailed in the Introduction (lines 69-83).

Regarding materials and methods: Subchapter 2.1. Experimental protocol should be clearly described. Is there known methodology (from literature) for performing described experimental design ? If so, please add reference.  We have added a sentence and reference as suggested by the reviewer.

Subchapter 2.2. You mentioned that the sample spectrum was the result of  24 coadded spectra with a resolution of 4 cm-1. Why 24 ? Please specify number of samples during sampling period (5th instar, 6th instar, frass) and how many spectra did you record per sample ? In relation to the 24 coadded spectra, we have set up this as it produces the best spectra for this type of samples.  In relation with the other two points (number of samples and spectra) we have added this information in the new version as suggested by the reviewer.

Subchapter 2.3. This subchapter should also be clearly described. If I understood correctly, preprocessed MIR spectra (baseline and Savitzky-Golay) were further used for performing PCA and PLS-DA ? It would be easier for following if you describe, in one sentence, the reason of PCA performing and in one sentence why did you choose PLS-DA ? Applicability of PLS models developed using preprocessed spectra was estimated based on the standard error of cross-validation and coefficient of determination in cross validation. What about RPD (residual predictive deviation) in order to evaluate a predictive ability of a given model ? Please calculate RPD.  These are commonly used methods in spectroscopy.  We think that the reader can read the extensive available literature in the field about the algorithms, and pre-processing methods.  The PLS predicts a dummy value (type of waste).  The calculation of a RPD value does not make any sense in this type of analysis, as this is not a quantitative model.

Results and discussion: Caption of figure 1 should contain type of waste. It is unclear is it for soy or vegetable mixture waste. Also, you should add the word preprocessed (average preprocessed spectra). You did not present figures of MIR spectra for both type of waste. Why ? We did not observe any visual relevant differences in the spectra, this was the main reason of using the absorbance values and compare them to show the differences.

Paragraph describing ratio of the absorbance between specific groups (panel A and B) should be clearly described for each type of waste. Accordingly, caption of figure 2 should be improved.  We have improved the section and caption as suggested by the reviewer.

Descriptions of figures 3 and 4 should be extended. For instance, figure 3, ....the combined set of instar (5th and 6th) and frass samples. For which type of waste ? Figure 4: for which type of waste ? We have improved the section and caption as suggested by the reviewer.

Description of table 1: ....the waste stream diet.....(for which type of waste) ? I think the reviewer does not understand what we are predicting. We are predicting the type of waste using a dummy regression combined the samples from the 5th and 6th instar.  Therefore, the title of the Table is correct.

Conclusion should be rewritten in a more scientific manner. The last sentence in a conclusion is unclear. We have improved the section and caption as suggested by the reviewer.

Round 2

Reviewer 2 Report

Figure 1. For which type of waste ? Still unclear.

For the second waste stream it would be appropriate to write "data not shown".

Regarding Table 1. For not understanding what are you predicting: not my problem. The point is that you did not clearly describe the aim of the study.

In the text, you should refer to Table 1.

Author Response

Figure 1. For which type of waste ? Still unclear.  The average of both waste streams. We have incorporated this information.

For the second waste stream it would be appropriate to write "data not shown". We have added as suggested by the reviewer.

Regarding Table 1. For not understanding what are you predicting: not my problem. The point is that you did not clearly describe the aim of the study. We have improved the aims as suggested by the reviewer.

In the text, you should refer to Table 1. We have added into the text as suggested by the reviewer.